# AG129 Mice as a Comprehensive Model for the Experimental Assessment of Mosquito Vector Competence for Arboviruses

**DOI:** 10.3390/pathogens11080879

**Published:** 2022-08-03

**Authors:** Lívia V. R. Baldon, Silvana F. de Mendonça, Flávia V. Ferreira, Fernanda O. Rezende, Siad C. G. Amadou, Thiago H. J. F. Leite, Marcele N. Rocha, João T. Marques, Luciano A. Moreira, Alvaro G. A. Ferreira

**Affiliations:** 1Mosquitos Vetores: Endossimbiontes e Interação Patógeno-Vetor, Instituto René Rachou-Fiocruz, Belo Horizonte 30190-002, Brazil; livia.baldon@aluno.fiocruz.br (L.V.R.B.); smendonca@aluno.fiocruz.br (S.F.d.M.); fvianaferreira@gmail.com (F.V.F.); fernanda.rezende@fiocruz.br (F.O.R.); marcelebio@yahoo.com.br (M.N.R.); luciano.andrade@fiocruz.br (L.A.M.); 2Departamento de Bioquímica e Imunologia, Instituto de Ciências Biológicas, Universidade Federal de Minas Gerais, 6627-Pampulha, Belo Horizonte 31270-901, Brazil; gbadeguetchin@gmail.com (S.C.G.A.); thjfl21@gmail.com (T.H.J.F.L.); jtmarques2009@gmail.com (J.T.M.); 3Faculté des Sciences de la Vie, Université de Strasbourg, CNRS UPR9022, Inserm U1257, 67084 Strasbourg, France

**Keywords:** arbovirus, mice model, AG129, vector competence, virus transmission, vertebrate transmission model

## Abstract

Arboviruses (an acronym for “arthropod-borne virus”), such as dengue, yellow fever, Zika, and Chikungunya, are important human pathogens transmitted by mosquitoes. These viruses impose a growing burden on public health. Despite laboratory mice having been used for decades for understanding the basic biological phenomena of these viruses, it was only recently that researchers started to develop immunocompromised animals to study the pathogenesis of arboviruses and their transmission in a way that parallels natural cycles. Here, we show that the AG129 mouse (IFN α/β/γ R^−/−^) is a suitable and comprehensive vertebrate model for studying the mosquito vector competence for the major arboviruses of medical importance, namely the dengue virus (DENV), yellow fever virus (YFV), Zika virus (ZIKV), Mayaro virus (MAYV), and Chikungunya virus (CHIKV). We found that, after intraperitoneal injection, AG129 mice developed a transient viremia lasting several days, peaking on day two or three post infection, for all five arboviruses tested in this study. Furthermore, we found that the observed viremia was ample enough to infect *Aedes aegypti* during a blood meal from the AG129 infected mice. Finally, we demonstrated that infected mosquitoes could transmit each of the tested arboviruses back to naïve AG129 mice, completing a full transmission cycle of these vector-borne viruses. Together, our data show that A129 mice are a simple and comprehensive vertebrate model for studies of vector competence, as well as investigations into other aspects of mosquito biology that can affect virus–host interactions.

## 1. Introduction

Arthropod-borne viruses (or arboviruses) are responsible for several medically important emerging and re-emerging diseases and are a major burden on many health systems around the world [1,2,3]. Despite arboviruses, such as dengue virus (DENV) and yellow fever virus (YFV), having been a threat to human health for centuries, in recent decades, several arboviruses have emerged (or re-emerged), and in some cases, are expanding their geographic distribution [4,5,6,7]. These include the Zika virus (ZIKV), West Nile virus (WNV), St. Louis encephalitis virus (SLEV), Japanese encephalitis virus (JEV), Chikungunya virus (CHIKV), Mayaro virus (MAYV), and Oropouche virus (OROV) [5,8,9,10,11,12,13,14]. Notwithstanding the growing public health threat caused by these emerging arboviruses, there are no specific treatments or efficacious vaccines for most of them [15].

Although other types of transmission can occur (such as sexual transmission) [16], in most of the cases, the arboviruses are transmitted from an infectious vertebrate host (the amplifying host) to another vertebrate via an intermediary hematophagous arthropod (the vector), such as ticks, sandflies, midges, and mosquitoes [17]. For the arboviruses to prevail, an efficient transmission between susceptible hosts is required [18]. Accordingly, it is vital that the vertebrate host can develop viremia so that arthropod vectors acquire infectious viruses along with blood during feeding [19]. For most of the arboviruses transmitted by mosquitoes, the virus undergoes replication in the midgut after the feeding acquisition [20]. Then, the virus disseminates to secondary sites of replication, including the salivary glands, and is ultimately released into the mosquito’s salivary secretions, where it may be inoculated into the skin and cutaneous vasculature of the host during subsequent feeding [18]. This inherent ability of a particular mosquito species to transmit a particular virus is denominated as vector competence, which is also vital for arbovirus perpetuation [21].

In addition to human infection prevention research, many recent studies involve mitigation strategies which rely on blocking transmission, either by controlling vector populations, or developing vectors that are resistant to arbovirus infections [22]. All too often, this experimental research includes testing for vector competence of mosquito species such as *Aedes aegypti* and *Culex quinquefasciatus.* In general, these vector competence studies involve analysis of both mosquito infection rates and mosquito transmission rates after an infectious blood meal [23]. To facilitate the experimental setup, females are often provided blood through artificial membrane feeders, where a mix of blood and virus produced in cell culture is used [24]. Although the virus stocks are commonly prepared using mammalian cell cultures, such as Vero cells, the artificial feeding systems do not fully resemble the viremic host in the field. In addition, forced salivation is one of the most common assays used to experimentally study the potential of mosquito species to transmit arboviruses [25]. Again, this experimental approach does not thoroughly resemble the natural cycle of transmission in nature.

Although mice models are the leading organisms used for understanding the biology of different infectious diseases, until recently, wild-type mice were not used in mosquito vector research, mainly because immunocompetent mice are not susceptible to most arboviruses [26]. However, in the last two decades, several immunocompromised knockout mice strains have been developed and tested for arboviruses infection, namely MAVS^−/−^, STAT1^−/−^, STAT2^−/−^, IRF3^−/−^, IRF7^−/−^, IFN α, β, R^−/−^, IFN γ R^−/−^, and IFNα, β, γ R^−/−^. Even though IFNα, β, γ R^−/−^ is a very permissive strain to develop viremia, various studies have been using different knockout strains to study mosquito vector competence [27,28,29,30,31,32,33,34,35,36,37].

In this study, we use a single mice strain, the AG129 strain [38], which is a double-knockout mouse lacking both alpha/beta and gamma interferon receptors (IFN α, β, γ R^−/−^), to test the vector competence of five different arboviruses (DENV, ZIKV, YFV, CHIKV, and MAYV). Through intraperitoneal injection of the arboviruses, we successfully obtained AG129 mice viremia sufficient to allow mosquitoes *Aedes aegypti* to acquire infectious viruses, along with blood, during feeding. Furthermore, we observe that the mosquitoes become infected from feeding in viremic AG129 mice, for all five arboviruses tested. Notably, we also demonstrate that infected mosquitoes can transmit each of the arboviruses tested back to naive AG129 mice, in this way, completing the transmission cycle.

## 2. Results

### 2.1. AG129 Mice Are Susceptible to Infection with DENV, ZIKV, YFV, CHIKV, and MAYV

To determine whether the AG129 strain, a double-knockout mouse model lacking receptors for both type I (α, β) and type II (γ) interferons (IFNAR^−/−^ IFNGR^−/−^DKO) is suitable for mosquito vector competence studies analyzing infection and transmission of arboviruses, we first tested mouse survival rates after inoculation with five different arboviruses from two different families: *Flaviviridae* and *Togaviridae* (Figure 1). The five different viruses used in this study represent the major arboviral disease agents of concern in Central and South America. We used DENV and YFV, two re-emerging viruses of the flavivirus genus that are endemic and epidemic in tropical regions of Central and South America (Figure 1). We also tested two viruses recently introduced in the Americas: the ZIKV virus, from the flavivirus genus, and the CHIKV virus, which belongs to the *Alphavirus* genus (Figure 1). Moreover, we tested MAYV, a sylvatic arbovirus belonging to the *Togaviridae* family (*Alphavirus* genus), that is responsible for an increasing number of outbreaks in several countries of Central and South America (Figure 1).

Here, we inoculated AG129 mice with the different arboviruses through intraperitoneal (IP) injection and analyzed the survival rates for seven days (Figure 2A). For the three *Flaviviridae* viruses assessed here, DENV-1, YFV, and ZIKV, we observed no lethality during the first four days post infection (d.p.i.). (Figure 2B–D). Although no lethality until seven d.p.i. was observed when AG129 mice were inoculated with DENV-1, we found that ZIKV infected mice start to succumb at five d.p.i. (Figure 2B,C). Like DENV-1, low mortality rate was observed for animals inoculated with YFV. By seven d.p.i., we observed a mortality rate of 0% for DENV-1, 71% for ZIKV, and 14% for YFV (Figure 2B–D). In contrast, early and high mortality rates were observed for both alphavirus evaluated in this study, with CHIKV and MAYV showing 100% mortality by just three and four d.p.i., respectively (Figure 2E,F).

Our next goal was to investigate whether AG129 mice show viremia for the tested viruses, which provided us with a vertebrate animal model for infection using mosquitoes with arboviruses. As outlined (Figure 3A), to investigate the kinetics of blood viremia, we estimated the viral load daily by RT-qPCR. We found that, by two d.p.i., all animals already presented viral RNA in the blood for all tested viruses (Figure 3B–F). For the *Flaviridae* viruses, DENV-1, YFV, and ZIKV, we found that RNA levels increased from day two until day four, and then started to decrease at five d.p.i. continuously until day seven d.p.i. (Figure 3B–D). A limitation of our approach is that AG129 mice inoculated with the *Alphavirus* CHIKV and MAYV were consistently terminal, with almost all animals succumbing by day three post-infection. Nevertheless, we observed a great increase in the RNA levels present in the mice’s blood, from day one to day two, in all animals tested for both CHIKV and MAYV (Figure 3E,F), indicative of a potential source of mosquito infection during a blood meal. Together, these results show that AG129 juvenile animals are highly susceptible to DENV-1, ZIKV, YFV, CHIKV, and MAYV infection.

### 2.2. Viremic AG129 Mice Are Capable of Transmitting DENV, ZIKV, YFV, CHIKV, and MAYV to Mosquito Vectors

The observation that AG129 mice can develop high viremia levels in the first days after inoculation indicates that they could be used for vector competence studies (infection and transmission). To test this hypothesis, female *Ae. aegypti* mosquitoes were allowed to feed on viremic AG129 mice (Figure 4A). For the flaviviruses DENV-1, ZIKV, and YFV we used three week-old AG129 mice and mosquitoes were exposed to the viremic animals three days post virus inoculation. Whereas for CHIKV and MAYV we used 8 weeks old AG129 mice and mosquitoes were exposed to the viremic animals two days post virus inoculation.

Using this setup, we found that viremic AG129 mice were able to infect *Ae. aegypti* female mosquitoes with all three flavivirus viruses (Figure 4B–D). Regardless of the virus, we found high infection rates at eight d.p.i. (Figure 4B–D). For instance, we observed that ZIKV viremic mice were able to infect 97% of the mosquitoes that took a blood meal from them, followed by DENV-1 viremic mice with 74%, and YFV with 57% of mosquitoes becoming infected after feeding on viremic animals (Figure 4B–D). Notably, we also found high infection rates at eight d.p.i. for CHIKV and MAYV reaching 91% and 100% respectably (Figure 4E,F).

### 2.3. AG129 Mouse Is a Valid Model to Study Mosquito-to-Vertebrate Transmission of Arbovirus

Measuring the ability of a mosquito to transmit an arbovirus is important, not only for laboratory vector competence assays, but also for most virus–host interaction studies. Most experimental studies rely on indirect methods to estimate arboviral transmission, such as forced salivation, because of the lack of a simple vertebrate host to analyze natural transmission. Mice have been used before in these experiments, but not in a systematic manner for the completion of the transmission cycle for different arboviruses. Thus, we tested whether infected *Ae. aegypti* female mosquitoes were able to acquire arboviruses from infected AG129 mice and transmit them to naïve animals. First, we infected five to seven-day-old female mosquitoes with arboviruses by allowing them to obtain blood meals from viremic AG129 mice (Figure 5A). Then, 14 days post the infectious blood meal, we allowed the female mosquitoes to take a second blood meal from a naïve AG129 mouse. Five to ten female mosquitoes were exposed to an AG129 mouse (two to three weeks old), for 30 min. After blood feeding, all engorged mosquitoes were collected and assessed for arbovirus infection. Two to three days post blood feeding, the mice were anesthetized, and a blood sample was collected to test for the presence of viruses (Figure 5A). Using RT-qPCR to assess the presence of viruses in the exposed AG129 mice, we found that infected *Ae. aegypti* mosquitoes were able to transmit all three flavivirus viruses, DENV-1, ZIKV, and YFV, to juvenile AG129 mice (Figure 5B–D). We also assessed the infection status of AG129 mice bitten by CHIKV and MAYV infected mosquitoes. As shown in Figure 5E,F, AG129 mice exhibited high levels of CHIKV and MAYV.

Overall, we observed that AG129 mice exposed to bites from as few as three mosquitoes become infected. This occurrence was observed for all tested viruses, and viral RNA was detected from mouse blood as early as three days post-mosquito exposure, for DENV-1, ZIKV, and YFV (Figure 5B–D), and only two days after exposure for CHIKV and MAYV (Figure 5E–F). Together, these data demonstrate the successful analysis of a full arbovirus cycle of mouse-to-mosquito-to-mouse transmission, thus indicating that AG129 immunocompromised mice are a reliable vertebrate model to study vector competence.

## 3. Discussion

Regardless of artificial blood meals being a convenient proxy, they do not fully mimic the complexity and intricacy of blood feeding on a vertebrate host. A blood meal is the primary route through which mosquitoes acquire an arbovirus infection, and thus blood components or their metabolites can be putative host factors that may regulate the susceptibility of mosquitoes to arboviruses. The host blood is digested and metabolized within two to four days after the blood meal, where simultaneously, the viruses start to infect and replicate in the midgut epithelium cells. Accordingly, recent studies suggest that blood components or their metabolites could directly modulate the immune response, thus affecting the capacity of *Ae. aegypti* to become infected [39,40,41,42]. Moreover, the blood digestion by the female mosquito can also interfere with susceptibility to viral infection. Analyses using the GABAergic system have also demonstrated that ingestion of blood by mosquitoes resulted in a robust gamma-aminobutyric acid (GABA) production from glutamic acid derived from blood protein digestion, which can enhance arbovirus infection replication in mosquitoes [40]. Another sophisticated study, performed by Zhu et al., showed that the status of iron deficiency in host blood might contribute to the vectorial permissiveness to the dengue virus, thereby facilitating its spread by mosquitoes [41]. In summary, several lines of evidence indicate that the blood feeding process on a live vertebrate host can modulate a wide spectrum of arbovirus infections in the mosquito vector.

Here, to address these questions and overcome the artificial blood feeding contentions, we assess the worthiness of the AG129 mice strain to perform experimental vector competence studies. Mosquito vector competence experimental research usually comprises studies on the capacity of the female mosquitoes to acquire and maintain the virus, support replication, and ultimately produce and transmit virus particles with enough infectivity to cause infection in a vertebrate host. Here, we first investigated whether AG129 mice developed viremia for a wide range of arboviruses, namely: DENV, ZIKV, YFV, CHIKV, and MAYV. As shown in previous studies [43,44], our results using AG129 mice showed that inoculation with DENV resulted in high levels of viral RNA present in the blood, lasting several days and peaking on day three after inoculation. Similarly, our results with ZIKV virus are in concordance with other studies that report that AG129 animals are highly susceptible to infection, resulting in results in transient viremia, and that they start to succumb within five to eight days post inoculation [45,46], while not reaching 100% mortality until seven days after inoculation, as reported in previously studies using AG129 mice [45,46]. These results could be explained by variations in propagation methods, as well as lower titers used in this study for the mice inoculation. Although one of the main objectives of this work was to explore the possibility of using a single mouse strain for all arbovirus, alternatively, the less immunocompromised A129 strain (single mutant type-I interferon receptor deficient) could be used for the more lethal virus (ZIKV, CHIKV, and MAYV). Previously studies using the A129 strain showed that when subcutaneously challenged with ZIKV, mice are also highly susceptible to this arbovirus. Moreover, these studies showed that A129 mice developed high viremia levels and while mortality was high, it was observed that time points to death where around six days, indicating that this mouse strain is a suitable animal model for some arboviruses [31,47]. Additionally, in our YFV experiments, we observed that intraperitoneal inoculation of AG129 juvenile mice led to a transient viremia, these results being in concordance with previously studies where A129 mice yielded similar results [48].

Since few studies have used AG129 mice as a vertebrate animal model to study CHIKV and MAYV [49,50], we decided to assess the permissiveness to infection, viremia, and mortality rates for these two alphaviruses. Since CHIKV and MAYV inoculation by intraperitoneal injection in AG129 juvenile mice can lead to high mortality rates, we used adult mice (eight to nine weeks old) for inoculation with these two viruses. Even though inoculation of adult AG129 mice resulted in a rapid mortality of the AG129 mice, surprisingly, we observed that these animals developed a high viremia as early as two days after inoculation. One explanation for the high and rapid mortality rates observed in AG129 mice after inoculation with CHIKV, MAYV, and to a lesser extent, with Zika, could be because we used mammalian cell-derived virus particles, rather than particles derived from mosquito cells. The effect of the virus origin on spreading differences in mice has been demonstrated in previous studies using the West Nile virus. Boylan et al. demonstrated that, in the C57BL76J mice, mammalian cell-derived replicon particles spread from the site of inoculation significantly more rapidly than the particles derived from mosquito cells [51]. Despite the observed high mortality rates in ZIKV, CHIKV, and MAYV, the results summarized above demonstrate that AG129 mice are suitable for infections that can cause a transient viremia for a wide spectrum of arboviruses.

A fundamental question, which required additional experimental assays, was whether the observed viremia in AG129 mice was sufficient to infect the female mosquitoes through blood meals. It was, therefore, important to determine the viral load and infection rates of all arboviruses used in this study. When we exposed *Ae. aegypti* mosquitoes to viremic AG129 mice, and then individually scored the infection status of all females that took a blood meal, we observed a high infection rate across all arboviruses. Corroborating our results, previous DENV and ZIKV studies also demonstrated that viremic AG129 mice are capable of transmitting viruses and infecting *Ae. aegypti* mosquitoes through blood meals [36,52]. However, in this study we also reasoned that AG129 mice could be used for other arboviruses, in addition to the flavivirus. Indeed, our results with CHIKV and MAYV demonstrate that these immunocompromised mice, the AG129 strain, are suitable to transmit and consequently infect mosquito vectors after blood-feeding in viremic animals.

A critical aspect of the experimental studies to quantify mosquito vector competence is the evaluation of the transmission potential of the viruses back to the vertebrate host. Although several methods had previously been developed to assess the transmission potential of mosquito vectors, such as the capillary tube technique to capture saliva from infected mosquitoes, these indirect measures do not parallel the complexity of the natural transmission of the arbovirus to a vertebrate host. Several studies suggested that mosquito saliva is associated with modulation of the vertebrate host immune response [53,54,55]. For example, in vesicular stomatitis New Jersey virus (VSNJ) and *Aedes triseriatus* mosquitoes, Limensand et al. demonstrated that mosquito saliva released in the mouse during blood feeding can potentiate viral infection [56]. Here, we used the AG129 model as a vertebrate host to replicate the natural transmission cycle by allowing infected mosquitoes to take blood meals on naïve mice and then testing individual mice at later points for infection. Our approach shows that AG129 mice are suitable and comprehensive vertebrate models for studying arbovirus transmission by *Ae. aegypti* mosquitoes, since for all viruses tested, we observed mice infection after mosquito blood feeding.

Although our results show that AG129 mice are flexible host models to assess vector competence in laboratory experiments, a limitation to our approach could be the logistical constraints associated with the use of murine animals. For instance, the large numbers of mice needed for this type of studies, as well as the synchronization of mosquito and mouse age, can be a plausible limiting factor. Conversely, one critical goal of this study was to establish a single mouse strain that could be used for a wide range of arbovirus, thus reducing the logistical constraints involved in the production and use of murine models. In summary, our results demonstrate that AG129 mice are useful and flexible host models to experimentally assess mosquito vector competence, and they can be used to investigate many aspects of virus–host interactions.

## 4. Materials and Methods

### 4.1. Mice Lineages

In this study, we used AG129 mice (IFN α/β/γ R^−/−^), a double knockout immunocompromised lineage that lacks both types of interferon receptors, type I interferon (IFN α/β) and II IFN (IFN γ) [57]. AG129 mice were bred and maintained at the Animal Facility of the Instituto René Rachou, Fiocruz Minas. Experiments were approved by the Institutional Animal Care and Use Committee, Comissão de Ética no Uso de Animais da Fiocruz (CEUA) and performed according to institutional guidelines (license number LW-26-20).

### 4.2. Mosquito Lineages and Mosquito Rearing

In this study, we used *Aedes aegypti* mosquitoes from the Bangkok (BKK) laboratory strain. All mosquitoes were reared under insectarium-controlled conditions, 28 °C and 70–80% relative humidity, in a 12/12 h light/dark cycle. Eggs were placed in plastic trays containing two liters of filtered tap water, supplemented with fish food (Tetramin, Tetra) for hatching, and larvae were maintained at a density of 200 larvae per tray. After emerging, adults were kept in 30 cm × 30 cm × 30 cm BugDorm insect cages, where mosquitoes were fed with 10% sucrose solution ad libitum.

### 4.3. Virus Propagation and Titration

In this study we used DENV-1 strain (DENV-1/H. sapiens/Brazil/Contagem/MG/BRMV09/2015) isolated from human blood in Contagem, MG, Brazil, in 2015 [58]. The ZIKV strain used in these experiments (ZIKV/H. sapiens/Brazil/BRPE243/2015) was isolated in 2015 [59]. For YFV, we used the isolate YFV377H [60]. For CHIKV, we used the isolate CHIKV/H. sapiens/Brazil/NT16/2018, that was isolated from the serum of one patient in Niteroi City, RJ state, in 2018. For MAYV, we used the isolate TRVL 4645 [58,61]. DENV-1 and YFV were propagated in C6/36 *A. albopictus* cells. C6/36 cells were maintained on L15 medium, supplemented with 10% FBS (fetal bovine serum) and 1× Antibiotic-Antimycotic (Gibco), as described [36]. Cells were seeded to 70% confluence and infected at a multiplicity of infection (MOI) of 0.01 and maintained for six to nine days at 28 °C. For ZIKV propagation, a similar procedure was followed using Vero (monkey) cells that were maintained in DMEM medium, supplemented with 5% FBS (fetal bovine serum) and 1x Antibiotic-Antimycotic (Gibco). Vero cells were seeded to a confluence of 70–80%, infected with ZIKV at an MOI of 0.01, and maintained for six days in culture. CHIKV and MAYV was propagated in VERO cells (monkey), cultured in DMEM medium (Gibco) supplemented with 5% fetal bovine serum (FBS) (Gibco), for four days at 37 °C and 5% CO_2_. For all viruses, the supernatant was collected and clarified by centrifugation to generate virus stocks that were kept at −80 °C prior to use. Mock supernatants used as controls were prepared under the same procedure, without virus infection. Titration of DENV-1 was performed in BHK-21 cells, while ZIKV, YFV, CHIKV, and MAYV were titrated in Vero cells, using the plaque assay method to determine the viral titer. Titration was performed in six-well tissue culture plates. We allowed the virus to adsorb for 1 h at 37 °C, then an overlay of 2% in carboxymethyl cellulose (CMC) in DMEM with 2% FBS was added. Plates were incubated at 37 °C and 5% CO_2_ for 5 days. Then, formaldehyde was added, and the cells were covered with a crystal violet stain (70% water, 30% methanol, and 0.25% crystal violet) to visualize the plaques. Both arbovirus stocks and arbovirus from mice serum were titrated in cells in six-well tissue culture plates.

### 4.4. Mice Inoculation with Arbovirus

Arbovirus inoculation of AG129 mice was accomplished by intraperitoneal injection (IP). For DENV-1, ZIKV, and YFV, three to four-week-old animals were inoculated. Since young juvenile mice are more susceptible to virus infections, for the alphaviruses CHIKV and MAYV, we used mice eight to nine-week-old mice for inoculation with these two viruses. For DENV-1 and YFV, approximately 10^6^ p.f.u. were injected, whereas for ZIKV, CHIKV, and MAYV, approximately 10^5^ p.f.u. were injected. Following inoculation, the mice were visually monitored daily and scored for morbidity and mortality. For the AG129 mice viremia kinetics experiments, a blood sample of around 0.2 mL was collected for each time point from a single mouse. After each single collection, the mice were euthanized. The blood collection was performed in the tail vein of anesthetized mice. For all experiments, the mice were bred and kept during the inoculation experiments in a specific-pathogen-free facility at Isnstituto René Rachou. The mice were maintained in a temperature- and humidity-controlled facility on a 12 h light/dark cycle with food and water ad libitum. For all experiments, male AG129 mice were used.

### 4.5. Mosquito Infection with Arbovirus

Five to seven-day-old mosquito females were transferred into cylindrical containers fitted with nylon mesh (0.88 mm hole size) on top and starved through sugar-deprivation for 24 h prior to mice blood feeding. All mosquito infections were performed using AG129 mice. Infected AG129 mice were anaesthetized three days post infection (for DENV-1, ZIKV, and YFV) and two days post infection (for CHIK and MAYV), using ketamine/xylazine (80/8 mg per Kg^−1^). Subsequently, anaesthetized mice were placed on the top of the netting-covered containers with mosquito females. Unshaved mice were placed in a prone position, with the entire ventral surface and limbs available to the mosquitoes. Mosquito females were allowed to feed on mice for 30 min. After blood feeding, fully engorged females were selected. All engorged females were placed in a container covered with nylon mesh with a cotton pad soaked with 10% glucose solution and with a plastic cup with soaked paper on the bottom for egg laying. Mice fed mosquito females were harvested individually for RNA extraction at eight days post feeding.

### 4.6. Arbovirus Transmission from Mosquitoes to AG129 Mice

To test whether mosquitoes were able to transmit arboviruses to AG129 mice, we first fed five to seven-day-old female mosquitoes on viremic AG129 mice. Fourteen days after the infection, female mosquitoes were allowed to feed on three to four-week-old anaesthetized naive AG129 mice. Five to ten female mosquitoes were exposed to each AG129 mouse for 30 min and after blood feeding, fully engorged females were selected and harvested individually for RNA extraction for arbovirus quantification. Three to four days after the AG129 mice were exposed to the mosquitoes, blood was collected for arbovirus quantification by RT-qPCR.

### 4.7. RNA Extraction and RT-qPCR

RNA extraction from serum samples or whole mosquito samples was performed using the Trizol method reagent (Invitrogen, Carlsbad, CA, USA), as previously described [36]. Total RNA was extracted, according to the manufacturer’s instructions, with some modifications. Briefly, 50 μL of serum or whole mosquitoes were placed in a 1.5 mL Eppendorf tube, and then 200 μL of Trizol and two glass beads were added. Then the samples were grounded vigorously in a bead beater for 90 s. Next, the samples were left for 10 min at room temperature and then, 40 μL of chloroform was added to the tube and mixed vigorously for 30 s. After 10 min at room temperature, the samples were centrifuged at 12,000× *g* for 15 min at 4 °C. Then supernatant was mixed with an equal volume of chilled isopropanol by reversed mixing for 2 min and left overnight at −20 °C. Next, the samples were centrifuged at 12,000× *g* for 5 min at 4 °C. The sediment was washed with of 75% (*v*/*v*) ethanol and air-dried for about 10 min. The purified RNA was dissolved in 20 μL of RNase-free water and stored at −80 °C. The total RNA extracted was reverse transcribed using M-MLV reverse transcriptase (Promega, Madison, WI, USA) and using random primers for initiation. Negative controls were prepared following the same protocol, without adding the reverse transcriptase. All real-time PCR reactions were performed using the QuantStudio 12K Real-Time PCR System (Applied Biosystems, Foster City, CA, USA), and the amplifications were performed using the SYBR Green PCR Master Mix (Applied Biosystems—Life Technologies, Foster City, CA, USA). The final reaction volume was 10 µL. The thermal cycling conditions were composed of Hold Stage (fast ramp to 95 °C, hold 20 s); PCR Stage (40 cycles of 95 °C, hold 15 s, fast ramp to 60 °C, hold 60 s); and Melt Stage (fast ramp to 95 °C, hold 15 s, fast ramp to 60 °C, hold 1 min, slow ramp of 0.05 °C/s to 95 °C, hold 15 s). All real-time PCR reactions were carried out in triplicate. The relative quantification of gene expression was determined using the 2ΔCt method, as previously described [36,62,63]. The viral RNA load was expressed relative to the endogenous control housekeeping gene, RPL32, for *Ae. Aegypti* and RPL32 AG129 mice. For *Ae. Aegypti*, the RPL32 primers were: Forward: 5′-AGC CGC GTG TTG TAC TCT G-3′ and Reverse: 5′-ACT TCT TCG TCC GCT TCT TG-3′. For mice, the RPL32 primers were: Forward: 5′-GCT GCC ATC TGT TTT ACG G-3′ and Reverse: 5′-TGA CTG GTG CCT GAT GAA CT-3′. For DENV-1, the primers were: Forward: 5′-TCG GAA GCT TGC TTA ACG TAG-3′ and Reverse: 5′-TCC GTT GGT TGT TCA TCA GA-3′. For ZIKV, the primers were: Forward: 5′-TCA AAC GAA TGG CAG TCA GTG-3′ and Reverse: 5′-GCT TGT TGA AGT GGT GGG AG-3′. For YFV, the primers were: Forward: 5′-GCT AAT TGA GGT GYA TTG GTC TGC-3′ and Reverse: 5′-CTG CTA ATC GCT CAA MGA ACG-3′. For CHIKV, the primers were: Forward: 5′-AAG CTY CGC GTC CTT TAC CAA G-3′ and Reverse: 5′-CCA AAT TGT CCY GGT CTT CCT-3′. For MAYV, the primers were: Forward: 5′-AAG CTY CGC GTC CTT TAC CAA G-3′ and Reverse: 5′-CCA AAT TGT CCY GGT CTT CCT-3′.

## Figures and Tables

**Figure 1 pathogens-11-00879-f001:**
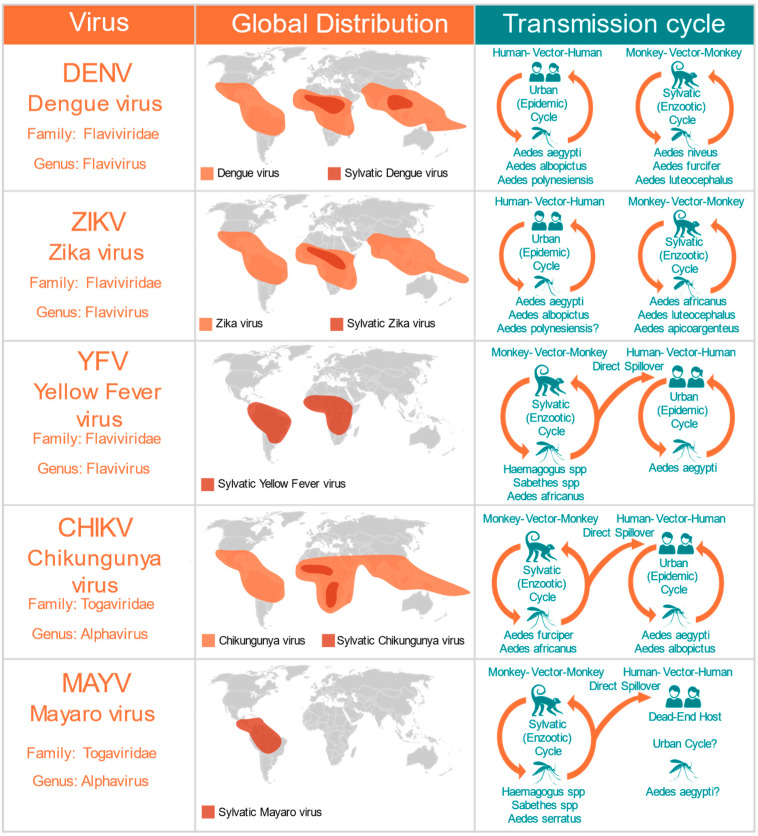
Medically important mosquito-borne arboviruses tested in the AG129 mice model. The global distribution includes the present the contemporary range, and both present and past outbreaks. The transmission cycles include the major vectors and the major reservoirs.

**Figure 2 pathogens-11-00879-f002:**
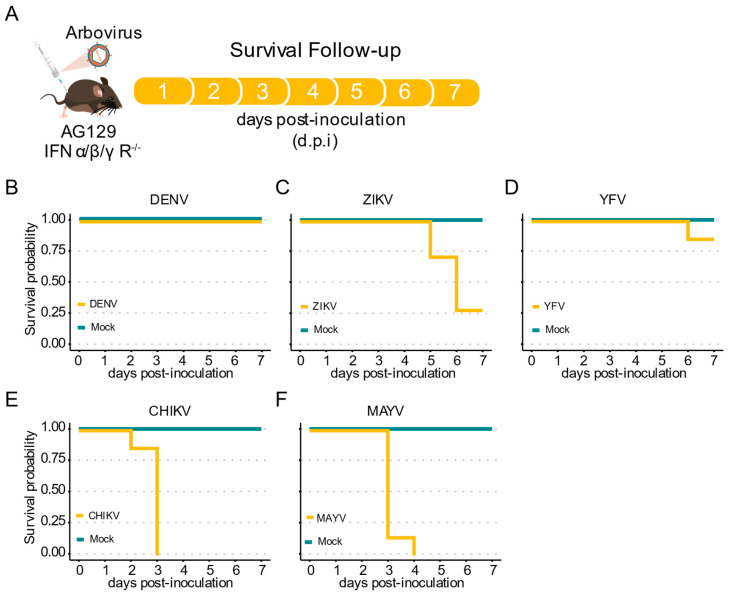
Mortality of immunocompromised AG129 (IFNα/β/γR^−^^/^^−^) mice after inoculation with arbovirus. (**A**) Scheme of the experimental design. Three or eight weeks-old AG129 mice were inoculated with arbovirus intraperitoneal (IP) injection performed in the lower right quadrant. (**B**–**F**) Survival probability (Kaplan–Meier plot) of AG129 mice inoculated with mock or viruses. Mice were monitored daily for seven days. For each virus seven mice were used. (**B**) Three-week-old mice inoculated with 10^6^ p.f.u. of DENV-1. (**C**) Three-week-old mice inoculated with 10^5^ p.f.u. of ZIKV. (**D**) Three-week-old mice inoculated with 10^6^ p.f.u. of YFV. (**E**) Eight-week-old mice inoculated with 10^5^ p.f.u. of CHIKV. (**F**) Eight-week-old mice inoculated with 10^5^ p.f.u. of MAYV.

**Figure 3 pathogens-11-00879-f003:**
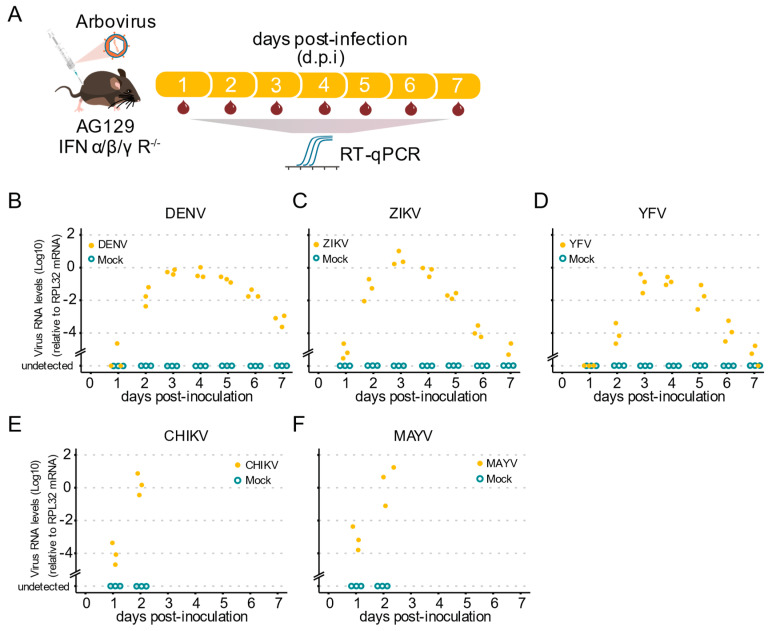
Arbovirus viremia in AG129 mice. (**A**) Scheme of the experimental design. Three or eight week-old AG129 mice were inoculated with arbovirus intraperitoneal (IP) injection performed in the lower right quadrant. Blood was collected every 24 h during 7 days for viral RNA quantification. Samples were tested individually by RT-qPCR. Each dot represents a blood sample from an individual mouse. For each time point, three different mice were sampled. Mice were monitored daily for seven days. Mice were euthanized after a single blood collection. (**B**–**F**) Virus RNA levels of blood samples from AG129 mice inoculated with mock or viruses. (**B**) Three-week-old mice inoculated with 10^6^ p.f.u. of DENV-1. (**C**) Three-week-old mice inoculated with 10^5^ p.f.u. of ZIKV. (**D**) Three-week-old mice inoculated with 10^6^ p.f.u. of YFV. (**E**) Eight-week-old mice inoculated with 10^5^ p.f.u. of CHIKV. (**F**) Eight-week-old mice inoculated with 10^5^ p.f.u. of MAYV.

**Figure 4 pathogens-11-00879-f004:**
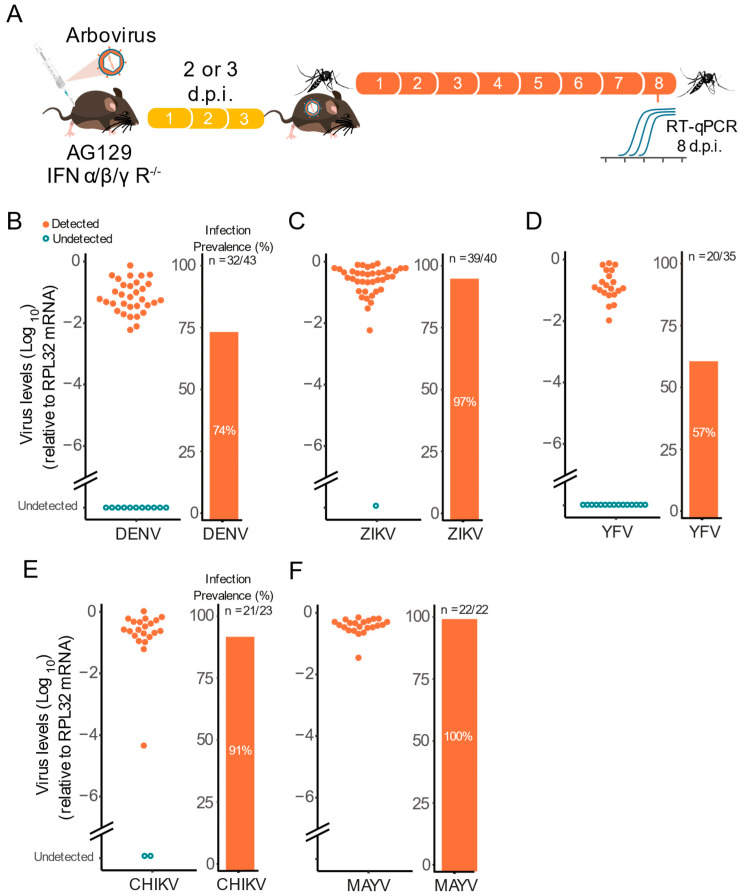
Viremic AG129 mice can transmit arbovirus to vector mosquitoes. (**A**) Scheme of the experimental design. Three or eight week-old AG129 mice were inoculated with arbovirus intraperitoneal (IP) injection performed in the lower right quadrant. After two or three days, mice were anaesthetized and then *Ae. aegypti* mosquitoes (five to seven-day-old females) were allowed to feed on virus-infected mice. Eight days post blood meal, mosquitoes were collected and tested individually for the presence of virus. (**B**) Three-week-old mice were inoculated with 10^6^ p.f.u. of DENV-1 and three days later, were exposed to mosquito blood-feeding. (**C**) Three-week-old mice were inoculated with 10^5^ p.f.u. of ZIKV and three days later, were exposed to mosquito blood-feeding. (**D**) Three-week-old mice were inoculated with 10^6^ p.f.u. of YFV and three days later, were exposed to mosquito blood-feeding. (**E**) Eight-week-old mice were inoculated with 10^5^ p.f.u. of CHIKV and two days later, were exposed to mosquito blood-feeding. (**F**) Eight-week-old mice were inoculated with 10^5^ p.f.u. of MAYV and two days later, were exposed to mosquito blood-feeding. In the infection prevalence bar graphs, “n” refers to the number of infected mosquitoes out of the total of mosquitoes tested.

**Figure 5 pathogens-11-00879-f005:**
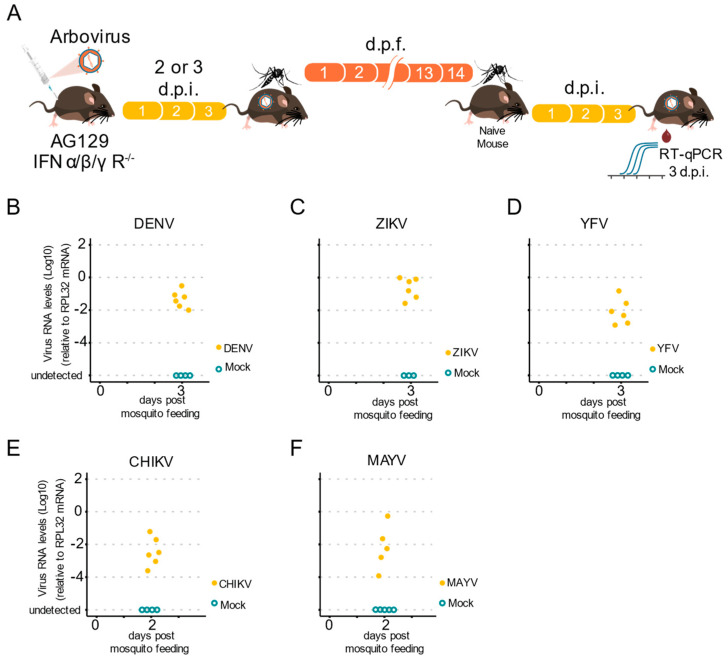
The AG129 mouse is a valid model to study mosquitoes-to-vertebrate transmission of arbovirus. (**A**) Scheme of the experimental design to test arbovirus transmission from mosquitoes to AG129 mice. Infected mosquitoes (that had taken an infectious blood meal from viremic AG129 mice 14 days earlier) were allowed to take blood meals in naïve AG129 mice for a second blood feeding. Three days later, the presence of arbovirus in the AG129 mice from which the mosquitos took the second blood meal, was detected by RT-qPCR. Each AG129 mouse was exposed to 5 infected mosquitoes for 30 min. Between two and five mosquitoes were able to complete a blood meal. Fully engorged mosquitoes were counted and collected to confirm the presence of viruses using RT-qPCR. Samples were tested individually by RT-qPCR. Each dot represents a blood sample from an individual mouse. (**B**) DENV-1 RNA levels in the mice after exposure to infected mosquitoes. (**C**) ZIKV RNA levels in the mice after exposure to infected mosquitoes. (**D**) YFV RNA levels in the mice after exposure to infected mosquitoes. (**E**) CHIKV RNA levels in the mice after exposure to infected mosquitoes. (**F**) MAYV RNA levels in the mice after exposure to infected mosquitoes.

## Data Availability

The data presented in this study are openly available in FigShare at doi:10.6084/m9.figshare.20419671.

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
