# Peer review of "AG129 Mice as a Comprehensive Model for the Experimental Assessment of Mosquito Vector Competence for Arboviruses"

_pathogens, 2022, doi:10.3390/pathogens11080879_

Round 1
Reviewer 1 Report
In the manuscript “AG129 mice as a comprehensive model for the experimental assessment of mosquito vector competence for arboviruses” the authors describe a useful evaluation of vector competence in a mammalian model as an alternative to artificial blood feeding models. While this is a mainly descriptive evaluation of the AG129 animal model and the ability to successfully act as a full transmission model, this manuscript is a welcome addition to the arboviral field. This is written clearly, and the conclusions are supported by the data. Suggestions for improvement are noted below.
Figure 1. Panels on global distribution could be made bolder colors for ease of viewing. Difficult to see light red over the grey background.
Line 114 (and throughout): DENV1 should be DENV-1
Line 115: d.p.i. is commonly “days post infection” rather than “days after infection.”
Overall consistency in either spelling numbers less than ten or using the numeral. For example, Lines 113-115, numbers are spelled and the numeral used in the same sentence.
Line 120: Alphavirus should be “alphaviruses”. The genus is not being referred to in this context so italics are not necessary.
Figure 2. The color scale between the lighter purple and grey is difficult to see.
Line 135. Period missing after the parentheses.
Line 199: should be “all three flaviviruses DENV-1, ZIKV, and YFV…” rather than “all three Flavivirus viruses”
Figure 5: “(B) MAYV” should be “(E) MAYV”
Lines 241-254: line 241 is an incomplete sentence and missing the first half of the statement. This paragraph requires editing as lines 247-254 are redundant of lines 220-226.
Line 267: “Studies with the other flavivirus, the YFV yielded similar results” should be “Studies with YFV yielded similar results”
Line 272: alphaviruses does not need italicized since the genus is not being referred to in this context, only viruses with the genus. Same in line 286, flavivirus does not need italicized and should be “flaviviruses” instead.
Line 309: “Nevertheless, the great feeding rates of the mosquitoes on the live vertebrate can contribute to compensate the above difficulties.” This sentence needs clarifying details and I’m not entirely sure what is meant and how it compensates.
Materials and Methods
4.1 Line 322 references injections of Oropouche virus, but that is not mentioned in the manuscript.
4.1 mentions mouse inoculations, as well as 4.4 (361-369). Please clarify these sections
Line 336: Chikungunya virus does not need italicized.
Line 337: what is the acute phase of “Niteroi, RJ disease”. Also the details on how the CHIKV was isolated are not necessary as the propagation of the virus for infections is most important and well described.
Section 4.3: the authors should mention why DENV-1 and YFV were propagated in C6/36 cells and ZIKV, MAYV, and CHIKV were propagated in Vero cells. Mammalian and insect cells have different complexities oligosaccharides added during the virion maturation that can have an effect on infection rates. Additionally, it has been shown that mammalian derived WNV spreads more rapidly than insect derived WNV (Boylan 2017 PLOS NTD). This discrepancy in the type of virus should be discussed as a possible limitation.
Section 4.4: it should be mentioned why older mice were used for MAYV and CHIKV.
Section 4.7 RNA extraction: What protocol was used for qRT-PCR? There are two reverse transcription descriptions in this section. Also why are gene expression calculations used for viral titer? One would not expect the viral replication to be a factor of gene expression. What were the control Cts vs treatment Cts, to compare to the housekeeping gene Cts to get the Delta Cts? A more conventional method would be determining RNA copies per uL using an RNA standard. What cycling conditions were used? Any references to the calculations or other manuscripts would be helpful in this section.
Reviewer 2 Report
This article addresses important methodology for the assessment of vector competence for several mosquito-borne viruses. There were some issues which, in my opinion, need addressing before it can be published.
MAJOR COMMENTS:
- Some results are unexpected. The ZIKV infection in AG129 mice is not 100% lethal, whereas for A129 mice (less immunocompromised) it is 100% lethal. The manuscript would be strengthened by comparisons with the A129 models for these arboviruses being included - as these are widely-used instead of AG129.
- The methodology needs further explanation. The age range of mice being 3 weeks for some and 8 weeks for others is significant, as 3 weeks is very young - so likely to be a lot more susceptible to infection. In line 150, the mice were euthanised after a single blood collection, so not sure why sequential bleeds were not collected to reduce animal usage? Line 368 contradicts this and implies it was the same animals. The route of blood collection, volume, etc should also be given. The animal methodology section (line 316) needs more details on gender, age, housing conditions, etc - as per standard ARRIVE guidelines for reporting in vivo studies. For line 374, more details on the mosquito feeding is required, e.g. the hole size of the netting. A picture or photo, even as supplementary information, would be extremely beneficial to visualise the set-up and how the animal is set-up for feeding (e.g. on side or back, is fur shaved, etc).
- The genome levels are displayed as relative to a house-keeping gene (RPL32). I am not sure why the authors used this approach is used, and not genome copies/ml or /mosquito as is generally used.
MINOR COMMENTS:
Line 19: change "have" to "having". Also in line 41.
Line 97: keep consistent whether using letters or latin symbols for alpha & gamma.
Figure 1, line 114 and others: "Flaviridae" should be "Flaviviridae" as per line 99.
Line 177: change term "tardily lethality".
Figure 2: survival follow-up shown as 8 days, but elsewhere goes to 7 days.
Line 222: expand GABA abbreviation on first use.
Figure 5: viral loads interesting, but would have been good to see some clinical data associated with these infections too (e.g. weight loss, survival).
Lines 250-254: section a repeat of lines 221-226.
Lines 263-268: the measurement of viral RNA indicates the presence of nucleic acid, but not live virus. PCR results are often detected for longer periods than live virus, so this section needs rewriting to address this.
Line 284 and elsewhere: "AG129 mice is capable" should be "AG129 mice are capable".
Round 2
Reviewer 2 Report
The authors have addressed the original comments previously raised.